# Identification of Potential Muscle Biomarkers in McArdle Disease: Insights from Muscle Proteome Analysis

**DOI:** 10.3390/ijms23094650

**Published:** 2022-04-22

**Authors:** Inés García-Consuegra, Sara Asensio-Peña, Rocío Garrido-Moraga, Tomàs Pinós, Cristina Domínguez-González, Alfredo Santalla, Gisela Nogales-Gadea, Pablo Serrano-Lorenzo, Antoni L. Andreu, Joaquín Arenas, José L. Zugaza, Alejandro Lucia, Miguel A. Martín

**Affiliations:** 1Mitochondrial and Neuromuscular Disorders Group, Hospital 12 de Octubre Health Research Institute (imas12), 28041 Madrid, Spain; inesgcg@hotmail.com (I.G.-C.); sarita.asensio@gmail.com (S.A.-P.); rociogarridorgm@gmail.com (R.G.-M.); cdgonzalez@salud.madrid.org (C.D.-G.); pserranolorenzo.imas12@h12o.es (P.S.-L.); joaquin.arenas@salud.madrid.org (J.A.); alejandro.lucia@universidadeuropea.es (A.L.); 2Centro de Investigación Biomédica en Red de Enfermedades Raras (CIBERER), 28029 Madrid, Spain; tomas.pinos@vhir.org; 3Mitochondrial and Neuromuscular Disorders Unit, Vall d’Hebron Institut de Recerca, Universitat Autònoma de Barcelona, 08193 Barcelona, Spain; 4Department of Computer and Sport Sciences, Universidad Pablo de Olavide, 41013 Sevilla, Spain; asanher@upo.es; 5Grup de Recerca en Malalties Neuromusculars i Neuropediàtriques, Department of Neurosciences, Institut d’Investigacio en Ciencies de la Salut Germans Trias i Pujol i Campus Can Ruti, Universitat Autònoma de Barcelona, 08916 Barcelona, Spain; gnogales@igtp.cat; 6EATRIS, European Infrastructure for Translational Medicine, 1019 Amsterdam, The Netherlands; toniandreu@eatris.eu; 7Achucarro Basque Center for Neuroscience, Science Park of the UPV/EHU, and Department of Genetics, Physical Anthropology, and Animal Physiology, Faculty of Science and Technology, UPV/EHU, 48940 Leioa, Spain; joseluis.zugaza@ehu.es; 8IKERBASQUE, Basque Foundation for Science, Plaza Euskadi 5, 48009 Bilbao, Spain; 9Faculty of Sport Sciences, Universidad Europea de Madrid, 28670 Madrid, Spain

**Keywords:** PYGM, myophosphorylase, proteomics, McArdle disease, GSDV, iTRAQ, skeletal muscle, metabolic myopathy, protein biomarkers

## Abstract

Glycogen storage disease type V (GSDV, McArdle disease) is a rare genetic myopathy caused by deficiency of the muscle isoform of glycogen phosphorylase (PYGM). This results in a block in the use of muscle glycogen as an energetic substrate, with subsequent exercise intolerance. The pathobiology of GSDV is still not fully understood, especially with regard to some features such as persistent muscle damage (i.e., even without prior exercise). We aimed at identifying potential muscle protein biomarkers of GSDV by analyzing the muscle proteome and the molecular networks associated with muscle dysfunction in these patients. Muscle biopsies from eight patients and eight healthy controls showing none of the features of McArdle disease, such as frequent contractures and persistent muscle damage, were studied by quantitative protein expression using isobaric tags for relative and absolute quantitation (iTRAQ) followed by artificial neuronal networks (ANNs) and topology analysis. Protein candidate validation was performed by Western blot. Several proteins predominantly involved in the process of muscle contraction and/or calcium homeostasis, such as myosin, sarcoplasmic/endoplasmic reticulum calcium ATPase 1, tropomyosin alpha-1 chain, troponin isoforms, and alpha-actinin-3, showed significantly lower expression levels in the muscle of GSDV patients. These proteins could be potential biomarkers of the persistent muscle damage in the absence of prior exertion reported in GSDV patients. Further studies are needed to elucidate the molecular mechanisms by which PYGM controls the expression of these proteins.

## 1. Introduction

Glycogen storage disease type V (GSDV) (OMIM#232600), also known as McArdle disease, is a rare autosomal recessive myopathy caused by biallelic pathogenic mutations in the *PYGM* gene [1] that result in deficiency of the skeletal muscle isoform of glycogen phosphorylase (or ‘myophosphorylase’, PYGM) [2]. GSDV has been reported to have an estimated prevalence of 1 in 100,000—350,000 people [2,3,4,5].

Because PYGM catalyzes the first rate-limiting step of glycogen metabolism (i.e., removal of terminal alpha-1,4-glycosidic bonds from the outer branches of this molecule to release glucose-1-phosphate), deficiency of this enzyme leads to a block in the use of glycogen as an energy source for muscle contraction [6]. Typical clinical features consist of muscle ‘crises’ of pain and fatigue, together with tachycardia during the first minutes of dynamic exercise (e.g., brisk walking) that are attenuated after 7–10 min have elapsed—the so-called ‘second wind’ phenomenon [2,7]. These episodes of early exercise intolerance are frequently accompanied by severe muscle contractures, potentially leading to rhabdomyolysis and subsequent myoglobinuria, as reflected by ‘dark urines’. Yet, another feature of the disease is a persistent status of muscle damage—(as reflected by very high circulating levels of intra-muscle proteins such as creatine kinase [CK]), even in the absence of physical exercise on the previous day(s) [8].

More than 170 pathogenic mutations (including missense, nonsense, in-frame, frameshift, and splicing variants) have been identified in the *PYGM* gene that cause McArdle disease [9,10]. Most of these mutations result in a total absence of PYGM activity [11] in the patients’ muscle tissue, except for two patients carrying deep-intronic mutations in compound heterozygosity that led to some residual (~1% of normal) enzyme activity, with subsequent amelioration in clinical phenotype [12]. There is no association between the *PYGM* genotype and disease phenotype, since patients with the same mutation(s) can show quite different degrees of clinical severity [13]. The pathobiology of GSDV is not fully understood, but it seems that the potential molecular consequences of the lack of glycogenolytic–derived ATP involve not only the expected energetic deficit for actin-myosin cross bridging, but also impairments in membrane pump function, excitation–contraction coupling, and sarcolemmal excitability [14].

In an attempt to identify potential muscle protein biomarkers and gain insight into the pathobiology of GSDV, we analyzed the targeted proteome in skeletal muscle biopsies obtained from both patients with histochemical and genetic diagnoses of GSDV and healthy controls. In this regard, since skeletal muscle is the only tissue that is clinically affected in all patients with GSDV, the control tissue was skeletal muscle biopsies from aged and sex-matched healthy individuals with normal PYGM activity and no signs of the typical features of McArdle disease, such as frequent exercise-induced contractures or persistent muscle damage in the absence of prior exertion.

We first used isobaric tags for relative and absolute quantitation (iTRAQ) analysis [15] to compare the muscle protein expression in patients vs. controls. This was followed by a systems biology network-based approach to identify key proteins involved in distinct pathways that could be related to the GSDV phenotype, such as the breakdown of muscle fibers, muscle contractures, and impairment in calcium homeostasis or in other physiological processes of the skeletal muscle. To this end, we applied the therapeutic performance mapping system (TPMS) machine learning-based technology, particularly by applying artificial neuronal networks (ANNs) that were ‘trained’ using the human protein network and drug-pathophysiology knowledge [16,17]. This technology has proven useful to identify non-obvious functional relationships for drug repurposing purposes [18,19,20] and biological data analysis and prioritization of proteins according to documented relationships with pathophysiological processes [21,22,23], especially in rare diseases or when sample sizes are limited. After the prioritization process, levels of the selected candidate proteins were analyzed using Western blot analyses.

Our results indicate that in addition to PYGM, myosin 1 (MYH-1), tropomyosin alpha-1 chain (TPM1), sarcoplasmic/endoplasmic reticulum calcium ATPase 1 (ATP2A1, also abbreviated as SERCA1), troponin isoforms (troponin I2, fast skeletal type [TNNI2] and troponin T3, fast skeletal type [TNNT3]), and alpha-actinin-3 (ACTN3) show a relationship with GSDV, with their levels reduced in the skeletal muscle tissue of GSDV patients with respect to healthy controls. Most of these proteins are involved in muscle contractures associated with altered calcium homeostasis.

## 2. Results

The main characteristics of the patients are shown in Table 1 and Table 2. Sex distribution (50% and 62.5% female in patients and controls; Chi-square test’s *p* = 0.625) and mean (± SD) age (patients: 38 ± 12 years; controls: 40 ± 9 years; Mann–Whitney’s U *p* = 0.711) did not differ between the two groups.

By quantitative proteome analysis of skeletal muscle biopsies obtained from the *Biceps brachii* or *Vastus lateralis* of eight GSDV patients and eight healthy controls using iTRAQ labeling followed by reversed-phase liquid chromatography-mass spectrometry (RP-LC-MS/MS), 178 proteins were identified. The patient and healthy control samples were separately pooled, and parallel double labeling was performed for each pool, resulting in two label values per group (113 and 115 for patients and 114 and 116 for controls); all values were referenced to the values of the 113 patients’ pool (Appendix A).

The peptide value distribution for each protein with peptide number >10 was compared between control and patient pool values, respectively, to obtain a total of 21 proteins with comparable control pool values, on the one hand, and differences between controls and patients, on the other (Table 3). These results were used to set a control/patient value ratio-based threshold, considering the mean of this value for these 21 proteins (= 1.675).

Next, we detected the most differentially expressed proteins by calculating the control/patient value ratio of the global data (i.e., for all 178 proteins detected [Appendix A regardless of the number of peptides measured) and identified 15 proteins with control/patient values ratio >1.676 (Table 4).

To allow analyses with TPMS technology, all data were mapped to 14 unique reviewed SwissProtKB entries (Table 4). Nine of the fourteen proteins exhibited at least a two-fold higher change in one of the control pools compared to the 113-labeled patient pool, which was used as a reference for labeling the rest of the proteins (indicated as bold values in Table 4).

The possible relationship between the most differentially expressed proteins (Table 4) and GSDV attending to their ‘molecular characterization’ was evaluated by means of ANNs (Table 5 and Appendix A show results considering GSDV as a whole or considering the different motives separately, respectively). Attending to the associated *p*-values, we sorted the ANN ranking score into four categories: ‘very strong’ (*p* < 0.01), ‘strong’ (*p* < 0.05), ‘medium-strong’ (*p* < 0.25), and ‘weak’ (*p* > 0.25) (Appendix A). Three proteins, ATP2A1, MYH1, and TPM1, showed a very strong relationship with GSDV (Table 5). These three proteins were part of the functional motif *elevated cytosolic calcium levels*, and specifically of the *persistent contraction of muscle cell* sub-motif for MYH1 and TPM1 (Appendix A). The troponin isoforms, TNNI2 and TNNT3, displayed a strong relationship with GSDV (Table 5) and were also effectors of the sub-motif *persistent contraction of muscle cells*. All the proteins were related to muscle structure and activity and showed a stronger relationship with GSDV definition than the enzyme PYGM, the defective protein in GSDV, which presented a medium-strong score with GSDV molecular characterization and was assigned as an effector of the glycogenolytic pathway.

The PDZ and LIM domain protein 7 (PDLIM7) and alpha-actinin-3 (ACTN3) showed a medium-strong score, and the rest of the evaluated proteins showed a low probability of being related to GSDV in a molecular-dependent manner (Table 5), according to the used molecular characterization.

After evaluating the relationship between the most differentially expressed proteins and each GSDV motif, as described by molecular characterization (Appendix A), we observed that the motif that exhibited the highest probability of a relationship with the available data was *elevated cytosolic calcium levels*, and particularly the submotif *persistent contraction of muscle cell*. Most of these proteins showed a strong or medium-strong probability of a relationship with the motif and submotif. In fact, for all the candidate proteins, the highest probability score was observed for *elevated cytosolic calcium levels*.

To further understand the intermolecular relationships identified by ANNs, we generated a protein interactome with the human protein network used for model construction and based on publicly available sources. This allowed us to identify the interaction between the most differentially expressed (‘candidate’) proteins and the effector proteins identified as important in GSDV molecular characterization (Table 5, Figure 1). Most of the candidates showed an interaction with effectors of the biological motives *elevated cytosolic calcium levels-persistent contraction of muscle cell* (Appendix A). However, two of the most differentially expressed proteins, the skeletal muscle isoform of the myosin regulatory light chain 2 (MYLPF) and PYGM, interacted with proteins belonging to the *modulation of alternative metabolic pathways for energy obtainment-increased glucose uptake* motif. In addition, according to the databases used (see topological analysis in the methods section), six of the most differentially expressed proteins (i.e., MYH1, ATP2A1, isoform 1 of four and a half LIM domains protein 1 (FHL1), four and a half LIM domains protein 3 (FHL3), aldose reductase (AKR1B1) and carboxymethylenebutenolidase homolog (CMBL)) did not directly interact with any of the GSDV effectors nor with any other most differentially expressed proteins (Table 5).

From the list of the most differentially expressed proteins, myosin light chain, phosphorylatable, fast skeletal muscle (MYLPF, Q96A32), and myosin binding protein (MYBPC2, Q14324) were not predicted to be related to GSDV (weak relationship in Table 5); however, they appeared highly connected to proteins within the *elevated cytosolic Ca^2+^ levels* motif (Figure 1), which could explain the medium-strong signal detected between these proteins and this motif (Appendix A) despite not being its effectors.

To validate the predictive results obtained using the ANN analysis strategy, skeletal muscle levels of a selected group of candidate proteins were also analyzed by Western blot in GSDV patients and healthy controls. We selected as candidates the most differentially expressed proteins that were classified in the ‘very strong’ and ‘strong’ categories according to their relationship with GSDV as a whole (Table 5): MYH1, ATP2A1, TPM1, TNNI2, and TNNT3. Besides these proteins, ACTN3 was also considered a candidate and analyzed despite being ranked in the medium-strong category, due to two relevant reasons: (i) it has been documented to interact with PYGM and implicated in altered muscle calcium handling in the *Actn3* deficient (knockout) mouse model [27], and (ii) at least in female patients, *ACTN3* genotypes might contribute to explaining individual variability in the phenotypic manifestation of this disorder [28,29]. We showed that the expression levels of all tested candidates (MYH1, ATP2A1, TPM1, TNNI2, TNNT3, and ACTN3) were significantly lower in patients than in controls (Figure 2 and Appendix A).

## 3. Discussion

GSDV is a metabolic myopathy typically characterized by exercise intolerance (i.e., muscle pain and early exertional fatigue). If the exercise stress is not reduced or halted, severe muscle contractures (beyond the usual baseline state of ‘persistent’ muscle contraction and damage) and eventual rhabdomyolysis might occur, which in some cases, could result in acute renal failure [4,30]. Although the knowledge of the molecular and pathophysiologic mechanisms of GSDV has improved during the last two decades, particularly with insights provided by clinical, molecular, or physiological studies in patients [4,14,24,31,32,33,34,35], as well as by studies in preclinical models [21,34,36,37,38,39,40], there is still no explanation (at least at the molecular level) for some recognized clinical features of the disease, notably the persistent muscle damage in the absence of previous physical exercise [8].

We therefore aimed at investigating in depth the muscle proteome and the molecular networks associated with muscle dysfunction in GSDV patients in an attempt to identify key muscle proteins as biomarkers that could help to understand the underlying molecular mechanisms of muscle dysfunction or damage. To the best of our knowledge, this question has not been explored previously. In a case-control design with muscle biopsies from histochemical and genetically proven GSDV patients and from healthy controls, we assessed quantitative protein expression using the iTRAQ technique and then performed a systems biology-based strategy, particularly applying ANNs and topology interactome networks to identify the best candidates. Our analysis suggested that some of the identified candidate proteins are related to GSDV disease predominantly through the motif *persistent contraction of muscle cells* due to elevated cytosolic calcium levels, with the proteins ACTN3, ATP2A1, MYH1, TNNT3, TPM1, and TNNI2 showing the highest predictive values among all the proteins evaluated. Furthermore, the topological analysis indicated that the candidate proteins identified in this study interact with proteins involved in the persistent contraction of muscle cells due to elevated cytosolic calcium levels and the modulation of alternative metabolic pathways for energy obtainment.

The levels of ACTN3, ATP2A1, MYH1, TNNT3, TPM1, and TNNI2 proteins were significantly lower in the skeletal muscle of patients compared with healthy controls. MYH1 is a skeletal muscle protein that, in coordination with actin, plays an essential role in the generation of energy for muscle contraction through ATP hydrolysis [41]. ATP2A1, the sarcoplasmic/endoplasmic reticulum calcium ATPase 1 (previously known as SERCA1), is a membrane protein that is responsible for the transport of calcium from the sarcoplasm back into the sarcoplasmic reticulum after each sarcomeric contraction, and whose function is dependent on the energy delivered by ATP hydrolysis. Likewise, ATPA21 contributes to the excitation/contraction balance involved in muscle activity [42]. A decrease in ATP2A1 levels would result in an impairment in the reuptake of calcium back into the sarcoplasmic reticulum after each contraction, with subsequent accumulation of this ion in the sarcoplasm and impairment of muscle fiber relaxation—that is, permanent muscle contraction and muscle contractures. Interestingly, besides the association of primary pathogenic genetic variants in the *ATP2A1* gene with Brody myopathy (OMIM#601003, a rare autosomal recessive disorder characterized by painless muscle cramping and exercise-induced impaired muscle relaxation) [43], other conditions linked with aging, neurodegeneration, and muscular dystrophy also depress ATP2A1 function with the potential to impair intracellular calcium homeostasis and contribute to muscle atrophy and weakness [42]. There is some controversy on how to assess calcium homeostasis in different human diseases since most research has been performed in murine models [44,45,46,47]. On the other hand, the stability of actin filaments in the muscle fibers is ensured by the function of tropomyosin (TPM1), which, in association with the troponin complex (TNNI2 and TNNT3), plays a key role in the regulation of calcium-dependent interactions during muscle contraction [48]. In addition, ACTN3 plays an important role in the stability of the contractile apparatus at the Z-line, where this protein cross-links and anchors actin filaments [49]. Therefore, our findings suggest that decreased expression of the aforementioned proteins in GSDV could be associated, at least in part, with the altered muscle contractile function and a probable alteration of muscle calcium kinetics in this disorder. On the other hand, PYGM could also be involved not only in energy generation from glycogen breakdown, but also in the O-linked β-N-acetylglucosamine (O-GlcNa)c post-translational modifications of some proteins [6,50]. In this effect, O-GlcNAcylation plays an important role in several skeletal muscle functions, including optimal modulation of calcium homeostasis in fibers [51,52].

Our study is limited by the small sample size, although we believe this is justifiable in the context of a rare condition such as McArdle disease. We also failed to collect all the samples from the same muscle, although the vast majority of samples corresponded to the *Biceps brachii,* and the proportion of muscle type (i.e., 6/2 for *Biceps brachii*/*Vastus lateralis*) was identical in patients and healthy controls. Importantly, our approach also lacked a comparison group of patients with similar features to those of McArdle disease, such as muscle contractures—although we are not aware of any neuromuscular condition where muscle contractures are as frequent or persistent as in McArdle disease—and therefore we cannot address if the detected differentially expressed proteins are primarily or secondarily regulated. In addition, it must be kept in mind that with regard to potential biomarkers of McArdle disease, our findings must be viewed as mechanistic—hopefully providing useful insights and framework for future research—rather than practical ones since muscle biopsies represent an invasive procedure and the molecular techniques used here are not easily available in any center. In turn, the method of RP-LC-MS/MS used here (or LC-MS/MS in general) is currently the most effective tool to discover and quantify the human proteome and represents an essential approach for the study of biological systems that is, in fact, routinely applied for diverse applications beyond relative or absolute proteome quantification, including biomarker discovery [53].

In conclusion, while keeping in mind the aforementioned limitations, our findings suggest some candidate proteins as potential biomarkers of GSDV. Our results provide a framework for future studies aimed at elucidating the molecular mechanisms by which PYGM controls the expression of the most relevant identified proteins.

## 4. Materials and Methods

### 4.1. Patient and Control Samples

Muscle biopsies from *Biceps brachii* (the muscle usually chosen by the pathologists of our center for diagnosis of neuromuscular conditions, including those cases of initial suspicion of McArdle disease) or *Vastus lateralis* (the muscle usually chosen in case of initial suspicion of lower-limb neuromuscular affectation) were rapidly frozen in liquid nitrogen and stored at −80 °C. Patients were diagnosed by identification of the presence of pathogenic mutations in homozygosity or compound heterozygosity in the *PYGM* gene as well as by muscle histochemical staining of the PYGM activity, as previously reported [9,54]. In two patients, we also performed an enzymatic assay of PYGM in muscle homogenates. We used 8 muscle biopsies obtained from GSDV patients of the neuromuscular disease department of Hospital 12 de Octubre (Madrid, Spain) (Table 1) and 8 age- and sex-matched control muscle biopsies obtained from controls recruited from the same center with initial suspicion of possible neuromuscular affectation that was ultimately free of neuromuscular disorders (including normal mitochondrial DNA), and showed normal histomorphology and histochemical muscle biopsy results, including normal staining for PYGM activity (thereby discarding the presence of GSDV) (Table 2). Of note, all the patients were instructed to refrain from performing physical exercise within 48 h prior to biopsy collection in order to prevent the potential confounding effects of exertional rhabdomyolysis and severe muscles contractures on subsequent protein expression levels (e.g., muscle proteins involved in calcium homeostasis).

### 4.2. Proteome Analysis

#### 4.2.1. Samples Preparation

Fifteen milligrams of each muscle sample were homogenized in lysis buffer (Tris Buffered Saline 1X Solution (TBS) pH 7.4, 1% SDS and protease inhibitors cocktail) using a glass potter on ice. Homogenate was boiled for 1 min, vortexed for 30 s (three times), and centrifuged for 1 min at 14,000 rpm. Total protein was quantified in the supernatant using a Micro BCA™ Protein Assay Kit (Thermo Scientific, Waltham, MA, USA) to set a final concentration of 50 μg/µL. Patient and control protein extracts were divided into two different pools to perform iTRAQ labeling followed by proteome analysis by RP-LC-MS/MS.

#### 4.2.2. In-gel Digestion and iTRAQ Labeling

The protein extracts (50–60 µg/pool) were resuspended in sample buffer (125 mM Tris-HCl pH 6.8, 4% SDS, 20% glycerol, 10% 2-mercaptoethanol, and 0.004% bromophenol blue), and loaded onto 1.2 cm wide wells of a conventional SDS-PAGE gel (0.75 mm-thick, 4% stacking, and 10% resolving). Electrophoresis was stopped when the front of the samples entered 3 mm into the resolving gel. The whole proteome of each wide well concentrated in the stacking/resolving gel interface was dyed with Coomassie brilliant blue, excised in cubes (0.75 mm × 2 mm × 2 mm), and placed into 0.5 mL microcentrifuge tubes [15]. The excised pieces were destained in acetonitrile:water (1:1), reduced with 10 mM dithiothreitol (DTT) 1 h at 56 °C, and alkylated with 50 mM iodoacetamide 1 h at room temperature in darkness. Finally, the samples were digested in situ with trypsin (sequencing grade) (Promega, Madison, WI, USA) as described by Shevchenko et al. [55]. The gel pieces were dehydrated by removing all liquid using sufficient acetonitrile and dried in a SpeedVac (Thermo Scientific, Waltham, MA, USA). The dried gel pieces were re-swollen in 50 mM ammonium bicarbonate pH 8.8 with 60 ng/µL trypsin at 5:1 protein:trypsin (*w*/*w*), kept on ice for 2 h, and incubated 12 h at 37 °C. Digestion was stopped by adding 1% TFA. Whole supernatants were dried down and then desalted onto OMIX Pipette tips C18 (Agilent Technologies; Glostrup, Denmark).

For relative quantification, 50 µg of tryptic digested peptides were labeled using chemicals from the iTRAQ reagent 8plex Multi-plex kit (Applied Biosystems, Foster City, CA, USA), essentially as described elsewhere [56]. We performed parallel double labeling using reagents 113 and 115 for patient pools and reagents 114 and 116 for controls pools (Appendix A shows the numerical values for each individual labeling [pools 113 to 116] referenced to the values of the 113 patients pool). Briefly, tryptic peptides were dissolved in 0.5 M triethylammonium bicarbonate (TEAB) pH 8, followed by peptide labeling. Each iTRAQ reagent was dissolved in 50 μL of isopropanol and added to the respective peptide mixture, and then incubated at room temperature for two hours. Labeling reaction was stopped by the addition of 0.1% formic acid. Whole supernatants were dried down, and the four samples were mixed to obtain the “4plex-labeled mixture”. The mixture was desalted onto OASIS HLB Extraction Cartridges (Waters Corporation; Milford, MA, USA) until RP-LC-MS/MS analysis.

#### 4.2.3. Protein Identification and Quantification

The desalted 4plex-labeled mixture was dried, resuspended in 10 µL of 0.1% formic acid, and analyzed by RP-LC-MS/MS in an Easy-nLC II system coupled to an ion trap LTQ-Orbitrap-Velos-Pro hybrid mass spectrometer (Thermo Scientific, Waltham, MA, USA). The peptides were concentrated by reverse phase chromatography using a 0.1 mm × 20 mm C18 RP precolumn (Thermo Scientific, Waltham, MA, USA) and then separated using a 0.075 mm × 250 mm C18 RP column (Thermo Scientific, Waltham, MA, USA) operating at 0.3 μL/min. Peptides were eluted using a 240 min dual gradient from 5 to 25% solvent B in 180 min followed by a gradient from 25 to 40% solvent B over 240 min (Solvent A: 0.1% formic acid in water, solvent B: 0.1% formic acid, 80% acetonitrile in water). Electrospray ionization, EIS, was conducted using a Nano-bore emitters Stainless Steel ID 30 μm (Proxeon Byosystems; Odense, Denmark) interface [57]. The instrument method consisted of a data-dependent top-20 experiment with an Orbitrap MS1 scan at a resolution (m/Δm) of 30,000 followed by either twenty high-energy collision dissociation (HCD) MS/MS mass-analyzed in the Orbitrap at 7500 (Δ m/m) resolution. MS2 experiments were performed using HCD to generate high resolution and high mass accuracy MS2 spectra. The minimum MS signal for triggering MS/MS was set to 500. The lock mass option was enabled for both the MS and MS/MS mode, and the polydimethylcyclosiloxane ions (protonated (C_2_H_6_OSi)_6_); *m*/*z* 445.120025) were used for internal recalibration of the mass spectra. Peptides were detected in survey scans from 400 to 1600 amu (1 μscan) using an isolation width of 2 u (in mass-to-charge ratio units), normalized collision energy of 40% for HCD fragmentation, and dynamic exclusion applied during 30 s periods. Precursors of unknown or +1 charge state were rejected.

#### 4.2.4. Data Analysis

Peptide identification from raw data was carried out using the SEQUEST algorithm (Proteome Discoverer 1.4, Thermo Scientific, Waltham, MA, USA). Database search was performed against uniprot-Homo.fasta. The following constraints were used for the searches: tryptic cleavage after Arg and Lys, up to two missed cleavage sites, and tolerances of 10 ppm for precursor ions and 0.05 Da for MS/MS fragment ions, and the searches were performed allowing optional Met oxidation, Cys carbamidomethylation, and iTRAQ reagent labeling at the N-terminus and lysine residues. Search against decoy database (integrated decoy approach) was conducted using a false discovery rate (FDR) < 0.01. All proteins were identified with at least two peptides with high confidence. Quantitation of iTRAQ labeled peptides was performed with Proteome Discoverer 1.4 using a Workflow for processing raw files with HCD spectra for quantification (and identification). The Reporter Ions Quantifier node contains a specific quantification method for i-TRAQ 8plex (in Thermo Scientific Instruments). For the ratio calculations, we used Quan Value corrections, and for the Protein quantification, we considered protein groups for peptide uniqueness and used only unique peptides. Tolerances of 10 ppm for peak integration and the most confident centroid for the integration method were selected.

#### 4.2.5. Determination of the Differentially Expressed Proteins

According to the experimental protocol followed, for controls, we obtained two values per protein, referring to a patient value (i.e., 114/113 and 116/113); for patients, we obtained two values per protein, referring to a patient value (i.e., 113/113 and 115/113). Because 113-labeled patient values were referred to as the same value, 113/113 patient values are a constant = 1 (Appendix A). In order to obtain a non-arbitrary difference threshold, based on control/patient protein values ratio, we took advantage of the multiple measures obtained for each protein (i.e., the unique peptide measures) to compare the peptide distribution per protein between the control pool values (114/113 and 116/113) and the patient pool value with a distribution of values (115/113). To this end, we selected those proteins for which > 10 peptide values were available (number of proteins = 27) and compared the distributions of the peptide values of each protein between the pool values, applying different statistical methods (Student’s t-test, Wilcoxon rank-sum, or one-way ANOVA tests) and using the Benjamini-Hochberg FDR [58] for multi-test correction. This comparison was performed in a two-step process: (i) we compared the distribution of peptide values per protein between the two control pool values (114/113 and 116/113) and considered the lowest q-value obtained to ensure lack of differences; and (ii) for proteins with comparable controls, we compared the distribution of peptide values per protein between the patient (115/113) and the control pool values (114/113 and 116/113), respectively, considering the highest q-value to ensure the difference between patients and controls (Appendix A). Thereafter, we used the proteins with a peptide number > 10, with a comparable peptide value distribution between control pool values, as well as those peptide value distributions that differed between control and patient pool values, respectively, to establish a control/patient value ratio threshold using the mean of the values. According to these results (Table 3), the threshold was set as 1676. Then, the control/patient value ratio of all proteins (either with number of peptides >10 or ≤10) was calculated and used to identify the most differential proteins (i.e., those showing a control/patient value ratio >1.675).

### 4.3. Systems Biology-Based Analysis—TPMS Technology

A systems biology analysis was carried out from the data obtained from the differential protein analysis. The potential molecular relationship between the differentially expressed proteins in muscle from both patients and controls was evaluated by means of ANNs, following Therapeutic Performance Mapping System (TMPS) technology (Anaxomics Biotech, Barcelona, Spain), which applies supervised machine learning methods based on human protein functional networks to infer clinical and protein level knowledge [17,19]. This approach involves the generation of mathematical models based on human protein networks and drug-pathophysiological knowledge through the use of artificial intelligence techniques. TPMS models are protein-based, using SwissProtKB protein entries as the model basic unit. Thus, the most differentially expressed proteins were mapped to their corresponding reviewed SwissProtKB entry. Firstly, GSDV was characterized by reviewing indexed scientific publications in PubMed, following a protocol based on a structured search, as previously described for TPMS application [59]. We typified a ‘causative motive’ (i.e., mutation in the *PYGM* gene) and several ‘symptomatic motives’ (i.e., biological processes occurring in the skeletal muscle as a consequence of ATP deficiency and ADP and Pi accumulation) that were subdivided into sub-motives (Appendix A). Of note, GSDV as a whole and each of its motives were considered distinct biological processes. Thereafter, mathematical models were solved by ANNs––that is, supervised algorithms that identify relationships between the different nodes in the network and obtain predictive values to functionally relate pre-defined protein sets. Thus, ANN analysis yields a score for each evaluated differential protein and each biological process, based on the validations of the prediction capacity of the mathematical models towards known drugs and diseases [16,59], as described in databases. The higher the score, the stronger the predicted mechanistic relationship between the evaluated protein and the biological process. Each score is associated with a *p*-value that describes the probability of a result being a true positive one. With the aim of facilitating the understanding of the results, scores were classified into four categories: >91, ‘very strong’ (*p* < 0.01); >76, ‘strong’ (*p* = 0.01–0.05); 40–76, ‘medium-strong’ (*p* = 0.05–0.25); and <40, ‘weak’(*p* > 0.25) (Appendix A). Proteins categorized as ‘very strong’ and ‘strong’ were further analyzed by Western blot to validate this strategy.

#### Topology Analysis

TPMS technology-based models rely on the human biological network [16,17]. This network incorporates all the documented relationships between proteins that are available from dedicated public sources, specifically Kyoto Encyclopedia of Genes and Genomes (KEGG) [60], REACTOME [61], IntAct Molecular Interaction Database [62], Biological General Repository for Interaction Datasets (BioGRID) [63], Human Protein Reference Database (HDPR) [64], and Molecular Interaction Database (MINT) [65]. The analysis of the network was focused on the area around the most differentially expressed proteins and the effector proteins identified as relevant in GSDV (McArdle disease) pathophysiology. The network was created using Cytoscape version 3.0.0. [26] (available from https://cytoscape.org/ (accessed on 15 February 2021)).

### 4.4. Western Blot of Candidate Proteins

Skeletal muscle samples were homogenized in five volumes of phosphate-buffered saline buffer (PBS) pH 7.4 1% SDS and then centrifuged at 14,000 rpm for 1 min. Protein concentrations in the supernatant were measured using a DC Kit (Bio Rad), and samples were diluted in loading buffer (12 mM Tris–HCl, pH 6.8, 0.4% SDS, 5% glycerol, 140 mM β-mercaptoethanol, and 0.02% bromophenol blue). Protein samples (20 µg) were separated on SDS-PAGE (4–20% gradient gels) and analyzed by immunoblot with specific anti-human antibodies: α-PYGM 1: 500 (Sigma, HPA056003), α-PYGL 1:250 (Sigma, HPA000962), α-PYGB 1:250 (Sigma, HPA031067), α-MYH1 1:500 (Abcam, ab190605), α-TNNT3 1:2000 (Abcam, ab175058), α-TPM1 1:500 (Abcam, ab109505), α-TNNI2 1:10000 (Abcam, ab183508), α-SERCA1 1:10000 (Abcam, ab105172), α-ACTN3 1:1000 (Abcam, ab68204), α-βTubulin 1:5000 (Abcam, ab10742), and then with a corresponding HRP-conjugated secondary antibody 1:1000–1:5000 (ab97023 from Abcam or G-212334 from Molecular Probes). Immunoreactive bands were visualized using ECL Clarity Max (Bio-Rad Laboratories; Hercules, CA, USA). The light-emitting bands were detected with ImageQuantTM LAS 4000 (GE-Healthcare; Amersham, UK). The resulting band intensities were quantitated using ImageJ 1.38 software (National Institutes of Health, Bethesda, MD, USA). To minimize the variability due to experimental conditions, we performed electrophoresis in two parallel gels that were running in the same electrophoretic cuvette and conditions. We also carried out the transfer of the separated products to the membranes, as well as the incubation with primary and secondary antibodies using the same conditions for the two gels. Finally, the exposure of membranes was conducted consecutively. Immunoblot analysis was performed in patients and controls of which muscle biopsy remanent after proteomic analysis were available (i.e., 7 patient samples since no muscle biopsy remanent from patient number 4 in Table 1 was available and 6 controls due to no muscle biopsy remanent were available for two controls). 

Mann–Whitney statistical test was used to compare control and patient protein levels, and *p*-values ≤ 0.05 were considered statistically significant.

## Figures and Tables

**Figure 1 ijms-23-04650-f001:**
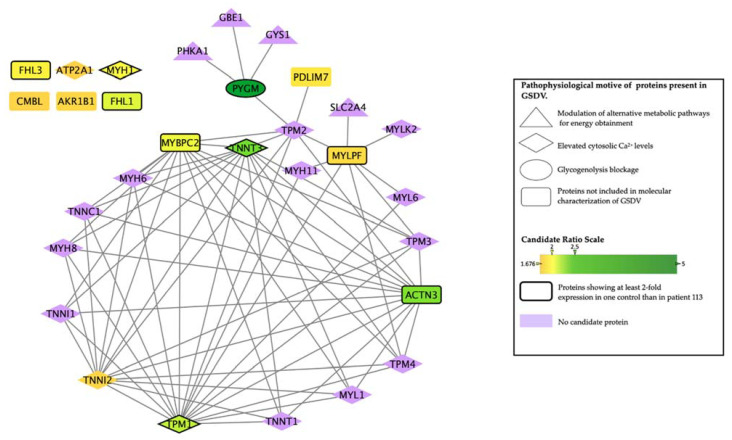
Interactome network map between the most differentially expressed proteins and the genes of GSDV effectors. The figure shows the most differentially expressed proteins and their internal relationships, as well as the relationships with the GSDV effectors. Symbols indicate whether the protein is included as an effector of each GSDV pathophysiological motif. The most differentially expressed proteins are highlighted according to a colored gradient showing the control/patient values ratio. Proteins with expression levels at least two-fold higher in one control than in patient 113 (i.e., 114/113 or 116/113 values) are marked by a thick border in the corresponding symbol. GSDV effectors not detected within the most differentially expressed proteins are depicted in purple color. Network built using TPMS human protein network [16,17] and visualized using Cytoscape version 3.0.0. [26].

**Figure 2 ijms-23-04650-f002:**
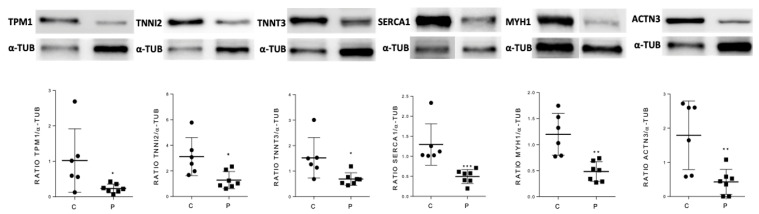
Western blot analysis in skeletal muscle tissue of the candidate proteins that were classified in the ‘very strong’ and ‘strong’ categories in the ‘molecular characterization’ of GSDV by means of artificial neural network analysis. Upper panel: Representative blots from GSDV patients and healthy controls (all patients’ and controls’ Western blots for each candidate protein displayed in Appendix A); alpha-tubulin was used as loading control. Bottom panel: Protein levels in GSDV patients (P, *n* = 7) compared to healthy controls (C, *n* = 6). * *p* < 0.05. ** *p*< 0.01,*** *p* < 0.001 using Mann–Whitney test. Each dot represents the mean of two quantifications (i.e., two technical replicates). Dispersion lines indicate mean ± SD.

**Table 1 ijms-23-04650-t001:** Demographic and disease characteristics of the GSDV patients.

Patient	*PYGM*	*PYGM*	Age	Sex	CK (U/L)	Severity Class ^2^	Muscle Used for Biopsy	PYGM	PYGM
	Allele 1 ^1^	Allele 2 ^1^	(Years)					Stain (Muscle)	Activity (Muscle)
1	c.148C>T	c.148C>T	19	F	1250	2	*Biceps brachii*	Absent	NR
	*p*.(R50*)	*p*.(R50*)							
2	c.148C>T	c.1366G>A	34	F	969	2	*Biceps brachii*	Absent	NR
	*p*.(R50*)	*p*.(V457M)							
3	c.2262delA	c.2262delA	32	F	500	1	*Vastus lateralis*	Absent	NR
	*p*.(K754Nfs*)	*p*.(K754Nfs*)							
4	c.148C>T	c.148C>T	24	M	533	2	*Vastus lateralis*	Absent	Absent
	*p*.(R50*)	*p*.(R50*)							
5	c.148C>T	c.613G>A	52	F	2328	2	*Biceps brachii*	Absent	NR
	*p*.(R50*)	*p*.(G205S)							
6	c.148C>T	c.2111C>T	48	M	4889	2	*Biceps brachii*	Absent	NR
	*p*.(R50*)	*p*.(A704V)							
7	c.148C>T	c.347T>C	55	M	1330	2	*Biceps brachii*	Absent	NR
	*p*.(R50*)	*p*.(L116P)							
8	c.2392T>C	c.2392T>C	43	M	1550	2	*Biceps brachii*	Absent	Absent
	*p*.(W798R)	*p*.(W798R)							

^1^ PYGM reference sequence: NM_005609.4; ^2^ as determined with the most commonly used phenotype severity scale for GSDV, the so-called ‘Martinuzzi scale’ (ranging from 0 [lowest] to 3 [highest]) for this disease [24]; where: 0 = asymptomatic or virtually asymptomatic (mild exercise intolerance, but no functional limitation in any daily life activity); 1 = exercise intolerance, contractures, myalgia, and limitation of acute strenuous exercise, and occasionally in daily life activities; no record of myoglobinuria, no muscle wasting or weakness; 2 = same as 1, plus recurrent exertional myoglobinuria, moderate restriction in exercise, and limitation in daily life activities; 3 = same as 2, plus fixed muscle weakness, with or without wasting and severe limitations on exercise and most daily life activities.

**Table 2 ijms-23-04650-t002:** Main characteristics of the healthy control group.

Control	Age	Sex	CK	Muscle Used for Biopsy	PYGM Stain (Muscle)
	(Years)		(U/L)		
1	41	M	N.A.	*Biceps brachii*	Normal
2	27	F	<200	*Vastus lateralis*	Normal
3	35	F	<200	*Biceps brachii*	Normal
4	52	M	N.A.	*Biceps brachii*	Normal
5	35	M	<200	*Biceps brachii*	Normal
6	40	F	<200	*Biceps brachii*	Normal
7	56	F	<200	*Biceps brachii*	Normal
8	34	F	N.A.	*Vastus lateralis*	Normal

**Table 3 ijms-23-04650-t003:** Proteins with peptide number > 10, or with comparable peptide values’ distribution between control pool values, or with different peptide value distribution between control and GSDV patient pool values, respectively.

UniProt ID	Protein	Gene	Control/Patient Values Ratio	Controls (114/113 vs. 116/113)	FDR q-Values	
					Controls vs. Patients	
					115/113 vs. 114/113	115/113 vs. 116/113
P11217	Glycogen phosphorylase, muscle isoform	*PYGM*	4.527	0.23	5.99 × 10^−11^	5.99 × 10^−11^
P20929-2	Nebulin	*NEB*	1.597	0.25	7.31 × 10^−7^	1.97 × 10^−7^
Q14324	Myosin-binding protein C, fast-type	*MYPC2*	2.035	0.22	3.39 × 10^−5^	5.50 × 10^−6^
P14618-2	Pyruvate kinase, isoform-1	*PKM1*	1.379	0.03	1.58 × 10^−8^	1.01 × 10^−10^
P08237	ATP-dependent 6-phosphofructokinase, muscle type	*PFKM*	1.521	0.26	8.77 × 10^−5^	8.77 × 10^−5^
P04075	Fructose-bisphosphate aldolase A	*ALDOA*	1.626	0.02	2.80 × 10^−6^	5.35 × 10^−7^
Q8WZ42-11	Titin, isoform-11	*TTN 11*	1.492	0.05	1.01 × 10^−10^	1.48 × 10^−11^
Q08043	Alpha-actinin-3	*ACTN3*	2.265	0.26	1.91 × 10^−7^	2.92 × 10^−8^
P00558-2	Phosphoglycerate kinase 1, isoform 2	*PGK1*	1.578	0.04	3.39 × 10^−5^	3.39 × 10^−5^
O14983-2	Isoform SERCA1A sarcoplasmic/endoplasmic reticulum calcium ATPase 1	*ATP2A1*	1.680	0.26	2.66 × 10^−5^	4.24 × 10^−6^
O60662	Kelch-like protein 41	*KLHL41*	1.469	0.26	2.12 × 10^−5^	2.12 × 10^−5^
P54296	Myomesin-2	*MYOM2*	1.465	0.45	2.12 × 10^−5^	2.12 × 10^−5^
P13929	β-enolase	*ENO3*	1.542	0.03	3.39 × 10^−5^	5.50 × 10^−6^
P16615-2	Isoform 2-sarcoplasmic/endoplasmic reticulum calcium ATPase 2	*ATP2A2A*	1.363	0.24	7.75 × 10^−3^	1.74 × 10^−3^
P12882	Myosin-1	*MYH-1*	1.972	0.04	8.49 × 10^−10^	1.46 × 10^−10^
P04406-2	Isoform2-glyceraldehyde-3-phosphate dehydrogenase	*GAPDH*	1.422	0.04	2.29 × 10^−4^	2.29 × 10^−4^
P06576	ATP synthase subunit beta, mitochondrial	*ATP5F1B*	1.157	0.13	1.63 × 10^−2^	2.69 × 10^−3^
P25705	ATP synthase subunit alpha, mitochondrial	*ATP5F1A*	1.142	0.07	1.34 × 10^−2^	1.05 × 10^−2^
P17661	Desmin	*DES*	1.200	0.14	1.54 × 10^−3^	2.83 × 10^−5^
P06732	Creatine kinase M-type	*CKM*	1.517	0.05	6.43 × 10^−9^	1.12 × 10^−9^
Q14315-2	Isoform 2-filamin-C	*FLNC*	1.253	0.05	5.83 × 10^−5^	1.20 × 10^−5^
**Control/patient values ratio mean**			1.676			

The table also shows the control/patient value ratio (considering the mean between 114/113 and 116/113 values as ‘control value’ and 115/113 value as ‘patient value’), the lowest q-value obtained for comparison within controls, and the highest q-value obtained for comparison between controls and patients. Tests applied: Student’s *t*-test, Wilcoxon rank-sum, or one-way ANOVA.

**Table 4 ijms-23-04650-t004:** Most differentially expressed muscle proteins in GSDV vs. controls.

UniProt ID	115/113 Value	114/113 Value	116/113 Value	Control/Patient Value Ratio	SwissProtKB ID ^1^	Protein	Gene
P11217	0.924	**4.056**	**4.315**	4.527	**P11217**	Glycogen phosphorylase, muscle form	*PYGM*
H9KVA2	0.916	1.979	**2.233**	2.298	**P45378**	Troponin T, fast skeletal muscle	*TNNT3*
C9JZN9	1.079	**2.263**	**2.668**	2.286			
Q08043	0.933	**2.002**	**2.224**	2.265	**Q08043**	Alpha-actinin-3	*ACTN3*
P09493	0.987	1.955	**2.265**	2.137	**P09493**	Four and a half LIM domains protein 3	*FHL3*
Q13642-1	1.050	1.942	**2.408**	2.072	**Q13642**	Tropomyosin alpha-1 chain	*TPM1*
Q14324	0.979	1.790	**2.196**	2.035	**Q14324**	Isoform 1 of four and a half LIM domains protein 1	*FHL1*
P12882	1.018	1.863	**2.151**	1.972	**P12882**	Myosin-1	*MYH1*
Q13643	1.150	1.992	**2.410**	1.913	**Q13643**	Isoform 6 of PDZ and LIM domain protein 7	*PDLIM7*
Q9NR12-6	1.027	1.844	1.919	1.832	Q9NR12	Myosin-binding protein C, fast-type	*MYBPC2*
Q96A32	1.078	1.712	**2.079**	1.759	**Q96A32**	Isoform 1A of sarcoplasmic/endoplasmic reticulum calcium ATPase 1	*ATP2A1*
P15121	0.910	1.473	1.676	1.731	P15121	Myosin regulatory light chain 2, skeletal muscle isoform	*MYLPF*
A0A087WXS0	0.961	1.686	1.614	1.718	P48788	Troponin I, fast skeletal muscle	*TNNI2*
Q96DG6	0.942	1.606	1.592	1.697	Q96DG6	Carboxymethylenebutenolidase homolog	*CMBL*
O14983-2	1.030	1.758	1.705	1.680	O14983	Aldose reductase	*AKR1B1*

^1^ Proteins showing at least a two-fold change in one control pool compared with the patient 113 pool are in boldface. UniProt ID as mapped by the SEQUEST algorithm. Values in boldface indicate patient values with respect to 113-labeled control values > 2. SwissProtKB ID indicates the corresponding reviewed SwissProt KB identifier associated with each proteome result.

**Table 5 ijms-23-04650-t005:** Relationship between candidate proteins and GSDV described by molecular characterization.

Gene ^1^	Uniprot ID	ANN Score ^2^	Score Category ^3^	GSDV Effector and Motif ^4^	Interactor GSDV Effector Genes
*MYH* (*)	P12882	93	Very strong	Elevated cytosolic calcium levels	-
				Persistent contraction of muscle cell	
*ATP2A1* [25]	O14983	93	Very strong	Elevated cytosolic calcium levels	-
*TPM1* (*)	P09493	92	Very strong	Elevated cytosolic calcium levels	*MYH11*; *MYH6*; *MYH8*; *MYL1*; *MYL6*; *TNNC1*; *TNNI1*; *TNNI2*; *TNNT1*; *TNNT3*; *TPM2*; *TPM3*; *TPM4*
				Persistent contraction of muscle cell	
*TNNI2* (*)	P48788	85	Strong	Elevated cytosolic calcium levels	*MYH6*; *MYH8*; *MYL1*; *TNNC1*; *TNNI1*; *TNNT1*; *TNNT3*; *TPM1*; *TPM2*; *TPM3*; *TPM4*
				Persistent contraction of muscle cell	
*TNNT3* (*)	P45378	83	Strong	Elevated cytosolic calcium levels	*MYH6*; *MYH8*; *MYL1*; *TNNC1*; *TNNI1*; *TNNI2*; *TNNT1*; *TPM1*; *TPM2*; *TPM3*; *TPM4*
				Persistent contraction of muscle cell	
*PYGM* (**)	P11217	75	Medium-Strong	Glycogenolysis blockade	*GBE1* *; *GYS1* *; *PHKA1* *; *TPM2*
*PDLIM7*	Q9NR12	66	Medium-Strong	-	*TPM2*
*ACTN3*	Q08043	43	Medium-Strong	-	*MYH6*; *MYH8*; *MYL1*; *TNNC1*; *TNNI1*; *TNNI2*; *TNNT1*; *TNNT3*; *TPM1*; *TPM2*; *TPM3*; *TPM4*
*MYBPC2*	Q14324	38	Weak	-	*MYH6*; *MYH8*; *MYL1*; *TNNC1*; *TNNI1*; *TNNI2*; *TNNT1*; *TNNT3*; *TPM1*; *TPM2*; *TPM3*; *TPM4*
*FHL3*	Q13643	36	Weak	-	-
*FHL1*	Q13642	27	Weak	-	-
*MYLPF*	Q96A32	18	Weak	-	*MYH11*; *MYL6*; *MYLK2*; *SLC2A4* *; *TPM1*; *TPM2*; *TPM3*; *TPM4*
*AKR1B1*	P15121	17	Weak	-	-
*CMBL*	Q96DG6	10	Weak	-	-

^1^ Superscript near the gene indicates the source from which the information for column ‘GSDV effector and motif’ was obtained: (*) *Molecular Cell Biology 4th Edition*, Textbook, ISBN-13, 978-0-7167-3706-3; (**) OMIM#232600, Online Mendelian Inherited in Man, https://omim.org (accessed on 15 February 2021); ^2^ ranking scores for the probability of relationship with the whole characterization of GSDV by means of ANNs; ^3^ category of the ANN ranking score (see Appendix A); ^4^ whether protein was previously described as implicated in GSDV or processes associated with muscle degradation.

## Data Availability

The datasets analyzed during the current study are available from the corresponding author upon reasonable request. The mass spectrometry proteomics data have been deposited to the ProteomeXchange Consortium via the PRIDE [66] partner repository with the dataset identifier PXD031605.

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
