# Peer review of "Identification of Potential Muscle Biomarkers in McArdle Disease: Insights from Muscle Proteome Analysis"

_ijms, 2022, doi:10.3390/ijms23094650_

Round 1

Reviewer 1 Report

Table 3. Relationship between candidate proteins and GSDV described by molecular characterization: it will look better to avoid blank lines where references are not present.

I suggest just removing the reference column while keeping the reference number near to the gene (for example MYH1 (n)). 

Author Response

Response to Reviewer 1 Comment

The manuscript has been revised by a native English speaker.

Table 3. Relationship between candidate proteins and GSDV described by molecular characterization: it will look better to avoid blank lines where references are not present.

Done. Thank you.

Please note that former Table 3 is now Table 5 in the revised manuscript (i.e., the new Table 5 is a fusion between the first submitted Table 3 and 4)

I suggest just removing the reference column while keeping the reference number near to the gene (for example MYH1 (n))

Done. Thank you. Please see above comment on new Table 5. 

Reviewer 2 Report

This is a muscle proteome anamysis in mcArdles disease, aiming to gain insight in the connection of differentially regulated proteins and clinical symptoms of the disease. McArdle shows quite different clinical symptoms than other glycogenoses with a higher amount of cramps and stiffness while other glycogenoses show rather flaccid paresis or work related symptoms. The approach of identifying biomarkers in the proteome level is a very insightful and well-known tool to identicy molecular mechanisms of a disease.

For my point of view, the flaw of this study design is a lack of knowledge about primary or secondary upregulation od proteines. Since McArdle patients have so many muscle cramps and stiffness, it would be no wonder if proteins chained to the process of muscle contraction and/or calcium homeostasis can also be secondarily upregulated. To distinguish a secondary effect from muscle contraction from a disease-immanent biomarker, it would be necessary to repeat the analysis in biopsies with non-specific cramps or e.g. spasticity.

Author Response

Response to Reviewer 2 Comments.

This is a muscle proteome anamysis in mcArdles disease, aiming to gain insight in the connection of differentially regulated proteins and clinical symptoms of the disease. McArdle shows quite different clinical symptoms than other glycogenoses with a higher amount of cramps and stiffness while other glycogenoses show rather flaccid paresis or work related symptoms. The approach of identifying biomarkers in the proteome level is a very insightful and well-known tool to identicy molecular mechanisms of a disease.

Comments much appreciated. Thank you.

For my point of view, the flaw of this study design is a lack of knowledge about primary or secondary upregulation od proteines. Since McArdle patients have so many muscle cramps and stiffness, it would be no wonder if proteins chained to the process of muscle contraction and/or calcium homeostasis can also be secondarily upregulated. To distinguish a secondary effect from muscle contraction from a disease-immanent biomarker, it would be necessary to repeat the analysis in biopsies with non-specific cramps or e.g. spasticity.

Thank you for this comment (please see additions/corrections [other than linguistic amendments]) highlighted in yellow in the revised manuscript).

The Reviewer is quite right when stating that the expression of some proteins (notably those involved in calcium homeostasis) could differ if the participant in question is suffering (or not) from exercise-induced ‘contractures’ [which we believe is a more correct term than ‘cramps’ in the context of McArdle disease] and subsequent stiffness. In this regard, as per routine protocol recommendations in our center when a neuromuscular condition is suspected, all the patients were told to refrain from performing physical exercise within the 2 days prior to biopsy collection in order to avoid potential confounders in the biopsy analyses. Please see revised manuscript, lines 381-385): 

Of note, all the patients were instructed to refrain from performing physical exercise within the 48 hours prior to biopsy collection in order to prevent the potential con-founding effects of exertional rhabdomyolysis and contractures on subsequent protein expression levels (e.g., muscle proteins involved in calcium homeostasis).

In addition, we have now provided data on serum CK levels (a well-accepted marker of muscle damage) in the patients (please see new Table 1) as well as in controls (please see new Table 2). As the Reviewer can see, except for patient #6 (where levels were a bit too high and yet actually not really so high in the context of McArdle disease) all the individual levels of CK were well within expected values at baseline (i.e., with no hard physical tasks in the previous 48 hours) for a McArdle patient, that is: 
mean of 1669 U/L and median of 1290 U/L in our patients (with all patients showing values above reference limits [>200 U/L] and 25% showing values above 2000 U/L). 

These values are similar (if not lower) compared to previous epidemiological data of CK levels at baseline in McArdle patients: above reference limits (>200 U/L) in nearly all patients, and above 2000 U/L in ~2 thirds of them, in the Spanish registry of patients (PMID: 29143597), or median of ~1300 to 1500 U/L in the European registry of patients (PMID: 33234167).

Importantly, indeed, one of the hallmarks of the disease (as mentioned in the revised manuscript, lines 65-69), is an ongoing status of persistent muscle contracture/stiffness and subsequent muscle damage even in the absence of previous physical exercise, as reflected by high CK levels at baseline. As such, the situation of all of our patients at the moment of the muscle proteome study was homogeneous. On the other hand, in case of exercise-induced contractures/stiffness and subsequent rhabdomyolysis beyond (or superimposed to) the aforementioned baseline situation of ongoing, persistent muscle damage, CK levels can go well above 5000 U/L and sometimes surpass 10000 U/L (which was not the situation shown by any of our patients).

On behalf of all co-authors, many thanks for this insightful review.

Reviewer 3 Report

The study by Garcia-Consuegra & al. presents a small study attempting to better describe proteomic changes occurring in the skeletal muscle of a cohort of 8 patients with McArdle disease. The hope of the manuscript is to obtain some clues that could help decipher pathogenic mechanisms explaining the muscle phenotype of these patients. Overall, the underlying goal of the manuscript is valuable as relatively little is available for this rare disorder and the use of primary tissues of patients is a strength of the study, which will help build translational effort between the bench and bedside. However, the current manuscript has multiple gaps (particularly in methodology and transparency), is somewhat difficult to follow and present associations that may not be strongly founded with the current data available. I have a few comments and suggestions for the authors to clarify some aspects of their manuscript.

Major comments

  1. In the optic of the manuscript being presented in a special issue of Biomarkers in Rare Diseases 2.0, please elaborate further on current biomarker research in McArdle disease, how the finding of this study may be used as biomarkers (screening, diagnostic, prognostic, etc) in the current research/therapeutic landscape. I am concerned that having to use muscle biopsy for biomarkers is rather invasive and not easily accessible.

  1. To follow up on point 1, there is minimal information about the patient characteristics in the manuscript apart from table 5 (which really should be table 1 – patients’ demographics should come at the beginning of the manuscript rather than the end). Any of the patient baseline characteristics, functional status, motor outcomes, laboratory data (ie CK) correlating with changes identified in the proteomic assay/levels change?

  1. Demographics of controls should be shown. The population should be compared to ensure that the population are well matched. The reason for work-up (biopsy) of the control should be mentioned to help the reader understand context and potential difference (or lack thereof) that may be observed.

  1. Please discuss in the manuscript the choice of muscle for the biopsy, predominant fiber types of biceps and vastus lateralis and proportion of each amongst the two groups (disease and control). I have concerns about different proportions between the two groups, knowing that biceps and vastus do not show the same profile of fibre type and that McArdle predominantly affect fast fibres. What steps were taken to minimize potential bias that may occur if the groups were not similar in this aspect?

  1. There is no supplementary table 1 available for revision, on which the very first part of the results including the proteomic results and a significant part of the method seems to rely on.

  1. I would highly encourage the authors to be more transparent throughout the manuscript (more to come in subsequent points). This would include showing all their data for proteomic. For example, dot plot of where all the protein fell with clearly defined their cut-off (1.5X more or less than control?) of what is a relevant change as well as how many proteins were deemed as unchanged. “21 showed significant differences between 109 patient and controls pools” is too vague. Additionally, table 1 only shows the protein, without their fold change nor p-values, making it hard to truly grasp the extent of change or even relevance of these changes.

  1. To follow-up with point 6, I would highly encourage that the proteomic set be deposited in a data depository. I do not see a clear statement that this would be performed.

  1. Line 114-147 – Very challenging to follow thought process. Keep in mind many readers will not familiar with the experimental framework used. I would encourage the authors to have a flow chart of their experimental designs.
    1. 114-116 – “calculating a control/patient pool ratio of the global data of each the aforementioned 21 proteins and the mean of the 21 values, setting the threshold at 1,675” is not clear.
    2. Please introduce ANNs, why their use would be advantageous to your study.
    3. Please describe where the motives come from. Were those hand-picked by the authors based on previous knowledge of the field? If so, a non-bias approach would be much preferred for this kind of study to limit bias, identify new mechanisms, or confirm previous data.
      1. Line 409 – 422 should probably be in the results as it would give context to the reader and help understand the direction taken by the authors.
    4. “we prioritized 14 protein candidates” – How did the authors make the decisions to take these proteins – What were the thought process?
    5. “least a two-fold higher change in one of the control pools than in the patient 113 pool” – What is patient 113 pool? The changes were only seen in one pool and not all of them? Why was this specific pool chosen rather than others or testing all of them?

  1. The data in the ANN scores for most, if not all the proteins as written in table 3 do not match the ANN score of Supp Table 3. For example, MYH1 as ANN score of 93 in table 3 and 91 in supp table 3. This actually changes the category of a few of these proteins, hence disrupt the reason for selection of a few targets for WB confirmation.

  1. First I want to commend the authors in providing the full WB, as this should be standard for publication. However, I have some concerns with the presentation. There is appalling variability in alpha-tubulin and loading altogether despite having known protein loaded. This is also seen in targets.
    1. I am afraid that “Upper panel: Representative blots” is quite a stretch – all the lanes should be shown to show variability.
    2. In the same way, a dot plot should be used to show the spread of the data.
    3. Pooling quantification of two membranes (WB A and WB B) is extremely risky and is arguably not a sound procedure considering that the two membranes are likely not exposed to the same environmental factors, conditions, etc.
    4. Many of the blots are overexposed (at least many lanes are), likely falling outside of the semi-quantitative range of chemilescence.
    5. With such variation on the blot, I am surprised to see a reasonable error bar. It is not clear to me whether the data represent mean + SD or SEM. This should be written in the figure caption along with the N-number for each experiment.
    6. Explain the reasoning for choosing Mann-Whitney test rather than Welsch or regular t-test?

  1. Figure 2. “Symbols indicate the pathophysiological motive in which the protein is involved. No candidates are in purple color, and candidates are highlighted according to a colored gradient depending of their probability of being candidate as indicated in the candidate ratio scale.” - This statement is very confusing especially how candidate is used throughout the manuscript.
    1. Candidates … depending of their probability of being candidate – What does this mean exactly? Candidate of what? Please refine or use an alternative word to ensure clarity.
    2. No candidate are in purple color – Please clarify.

  1. Statement: “our findings suggest that decreased expression of the aforementioned proteins in GSDV could explain the altered muscle contractile function and a probable alteration of muscle calcium kinetics in this disorder.” While this is true, reduction of protein does not necessarily entail functional deficits and functional experiments are required - The authors may consider discussing this aspect.

Minor comments

  1. How was the protein interactome obtained (software would be helpful)? This can be briefly mentioned in the results to help reader understand steps.

  1. Table 4 Define Interactor GSDV effector genes

  1. Statement 261-262 with comparison of ALS if perhaps not a fair and appropriate comparison given largely different pathogenic process and phenotype. ALS is highlighted by chronic denervation, muscle atrophy (and muscle weakness, rather than sustained muscle contraction), likely leading to reduced amount of these proteins. To my understanding, muscle atrophy is highlighted in the proteomic screen and as such the comparison is relatively weak.

Author Response

Response to Reviewer 3 Comments.

The manuscript has been revised by a native English speaker.

The study by Garcia-Consuegra & al. presents a small study attempting to better describe proteomic changes occurring in the skeletal muscle of a cohort of 8 patients with McArdle disease. The hope of the manuscript is to obtain some clues that could help decipher pathogenic mechanisms explaining the muscle phenotype of these patients. Overall, the underlying goal of the manuscript is valuable as relatively little is available for this rare disorder and the use of primary tissues of patients is a strength of the study, which will help build translational effort between the bench and bedside. However, the current manuscript has multiple gaps (particularly in methodology and transparency), is somewhat difficult to follow and present associations that may not be strongly founded with the current data available. I have a few comments and suggestions for the authors to clarify some aspects of their manuscript.

Comments appreciated. Thank you. 

Please see our point-by-point responses below. We have revised the manuscript in depth following the recommendations made by the Reviewer (please see additions/corrections [other than linguistic amendments]) highlighted in yellow in the revised manuscript).

Major comments

1. In the optic of the manuscript being presented in a special issue of Biomarkers in Rare Diseases 2.0, please elaborate further on current biomarker research in McArdle disease, how the finding of this study may be used as biomarkers (screening, diagnostic, prognostic, etc) in the current research/therapeutic landscape. I am concerned that having to use muscle biopsy for biomarkers is rather invasive and not easily accessible.

We agree with the Reviewer. Thanks in fact for raising this issue. Of note, when we got the official editorial invitation for this specific issue, we replied to the Editor describing the main scope of our paper and we were told that the manuscript fitted well within the issue. This being said, the Reviewer is quite right to point the issue of invasiveness. Please see additions at the end of the revised Discussion (lines 347-359), when we acknowledge the main limitations (as well as the main strengths) of our study.

2.     To follow up on point 1, there is minimal information about the patient characteristics in the manuscript apart from table 5 (which really should be table 1 – patients’ demographics should come at the beginning of the manuscript rather than the end). Any of the patient baseline characteristics, functional status, motor outcomes, laboratory data (ie CK) correlating with changes identified in the proteomic assay/levels change?

Former Table 5 has now become Table 1, which is now much more complete with the addition of three new columns (with information on patients’ CK levels, as well as on the specific muscle where the biopsy was taken, and the clinical severity at the moment of the study). Please see revised Table.
As for the potential correlation between (or effect of) CK/clinical severity and proteome results, if the Reviewer agrees we prefer not to make further analysis for the sake of simplicity (the paper is already complex) and due to the limited sample size (we would indeed run the risk of finding spurious correlations). Of note, as stated in the last response to Reviewer #2, CK levels were well within the range expected for a baseline situation in any McArdle patient based on epidemiological evidence  (PMID: 29143597) (PMID: 33234167). Importantly, indeed, one of the hallmarks of this disease (as mentioned in the revised manuscript, lines 65-69), is an ongoing status of ‘persistent muscle contraction/contracture’ and subsequent muscle damage in the absence of previous physical exercise. This is well reflected by high CK levels at baseline (that is, the situation of all of our patients at the moment of the muscle proteome study was a normal one). On the other hand, in the case of exercise-induced contractures/stiffness and subsequent rhabdomyolysis (which was not the situation shown by any of our patients), the CK levels of McArdle patients can go well above 5000 U/L (if not sometimes >10000 U/L).
On the other hand, all the patients belonged to one of the two most frequent severity classes (on a 0 to 3 scale) (PMID: 29143597), i.e., class 1 in one patient and class 2 in the rest of patients. Thus, the study population was relatively homogeneous in terms of phenotype manifestation. In addition, the two more ‘extreme’ classes [i.e., class 0 (= essentially asymptomatic) and class 3 (= all the disease features very marked + chronic muscle weakness)] were not represented.

3.     Demographics of controls should be shown. 

Done. Please see revised manuscript (new Table 2). 

The population should be compared to ensure that the population are well matched.

Done. Please see revised manuscript, lines 111-114 (similar sex distribution (p=0.614) and very similar age (p=0.711) in the two groups).

The reason for work-up (biopsy) of the control should be mentioned to help the reader understand context and potential difference (or lack thereof) that may be observed.

Done. Please see revised manuscript, lines 85-87. Indeed, since the only tissue that is clinically affected in McArdle patients is the skeletal muscle (with this condition being essentially a ‘pure myopathy’), the study control/reference had to be necessarily skeletal muscle from healthy age and sex-matched controls.

4. Please discuss in the manuscript the choice of muscle for the biopsy, predominant fiber types of biceps and vastus lateralis and proportion of each amongst the two groups (disease and control). 

Done. Please see revised manuscript, lines 367-370 (methods section). Biceps brachii is the usual muscle of choice in our center unless initial suspicion of specific lower-limb affectation.

I have concerns about different proportions between the two groups, knowing that biceps and vastus do not show the same profile of fibre type and that McArdle predominantly affect fast fibres. 

Sorry if we did not make it clear in the original submission. Yet, as we have now specified in the new Tables 1 and 2, except in two patients and two controls (i.e., same proportion in the two groups) all the biopsies were actually taken from the same muscle (biceps brachii), which is the usual muscle of choice for diagnostic biopsies in our center unless primary neuromuscular affectation of lower limbs is suspected.

On the other hand, although we are well aware of interesting preclinical data showing the effect of the muscle type on the molecular phenotype manifestation of McArdle disease as we have recently reviewed (e.g., the quadriceps and soleus muscles seem to be less histologically affected by disease progression than gastrocnemius, tibialis anterior or extensor digitorum longus) (see PMID: 35052414 for a recent review by our group), we have previously shown in patients with McArdle disease (PMID: 25432515) that: 
(1) McArdle disease does not affect skeletal muscle fibre type profiles in humans, with no difference in myosin heavy chain (MHC) isoform content between these patients and healthy controls for either vastus lateralis or biceps brachii, the two muscles studied here. 
(2) The MHC content is quite similar between the two muscles in patients, at least for the proportion of slow muscle fibers (i.e., vastus lateralis: MHC I, 33±19%; MHC IIa, 52±9%; and MHC IIx: 15±18%; biceps brachii:   MCHI, 33±14%; MHC IIa: 46±17%; and MHC IIx: 21±13%).

What steps were taken to minimize potential bias that may occur if the groups were not similar in this aspect?

Please see our above response. All the samples were taken from the same muscle (biceps brachii) in 6 of 8 subjects in both patients’ and control group, and the other muscle of choice was the same (vastus lateralis) in the two other patients and controls, respectively. 

5.     There is no supplementary table 1 available for revision, on which the very first part of the results including the proteomic results and a significant part of the method seems to rely on.

Our apologies, this must be an error of the online system beyond our control. All the suppl. files should not be visible to the Reviewer. 

In addition, we have changed the former Ms_Word format of Suppl. Table 1 to a Ms_Excel format showing the proteomic results.

6. I would highly encourage the authors to be more transparent throughout the manuscript (more to come in subsequent points). This would include showing all their data for proteomic. For example, dot plot of where all the protein fell with clearly defined their cut-off (1.5X more or less than control?) of what is a relevant change as well as how many proteins were deemed as unchanged. “21 showed significant differences between 109 patient and controls pools” is too vague. Additionally, table 1 only shows the protein, without their fold change nor p-values, making it hard to truly grasp the extent of change or even relevance of these changes.

As we indicated in the previous response #5, we have now replaced Word format by Ms_Excel in Supp. Table 1 in order to be totally transparent and show all the data for proteomic analyses. The legend of this Ms_Excel file is in the Ms_Word supplementary material that includes the rest of the suppl. tables. 
We have also revised in depth all steps of the analyses including tables and suppl. material (highlighted in different parts of the manuscript) to clearly define the cutoffs. 
We have introduced the analysis and data type used for statistical analyses in the Results section (lines 138-157), according to other paragraphs included through the Results and Methods section 4.2.5. We believe this should help to make the results clearer and more understandable (including the sentence “21 showed significant differences between patient and controls pools”).  

Please note that former Table 1 is now Table 3.  We have added more information to Table 3. Table 3 shows: i) proteins with peptide number >10, ii) peptide values distributions that were comparable between control and patient pool values, respectively, and (iii) peptide values distribution that differed between control and patient pool values, respectively; the control/patient values ratio (considering the mean between 114/113 and 116/113 values as ‘control value’ and 115/113 value as ‘patient value’); the lowest q-value obtained for comparison between controls; and the highest q-value obtained for comparison between controls and patients (see also methods section).

7.     To follow-up with point 6, I would highly encourage that the proteomic set be deposited in a data depository. I do not see a clear statement that this would be performed.

Done. We have submitted the dataset to ProteomeXChange via the PRIDE database. The data is currently private, and can only be accessed with a created reviewer’s single account (Username: [email protected]  Password: qI27ovFK), but it will become publicly accessed upon the first (online) publication the manuscript, if eventually accepted. Please see added sentences included in the revised manuscript, lines 590-591. 

1.     Line 114-147 – Very challenging to follow thought process. Keep in mind many readers will not familiar with the experimental framework used. I would encourage the authors to have a flow chart of their experimental designs. 

We hope that, thanks to the Reviewer’s insightful suggestions, the in-depth changes we have now incorporated throughout the manuscript (experimental procedures) have made the manuscript clearer and more understandable.

1.     114-116 – “calculating a control/patient pool ratio of the global data of each the aforementioned 21 proteins and the mean of the 21 values, setting the threshold at 1,675” is not clear.

Please see the modified section in the revised manuscript where we describe how the threshold is calculated (Methods section 4.2.5.).

2.     Please introduce ANNs, why their use would be advantageous to your study. 

We have included new text additions and references related to the application of our method (ANN through TPMS) to explain the benefits of use of ANNs in this context of a study as ours (lines 93-99 and lines 503-520).

3.     Please describe where the motives come from. Were those hand-picked by the authors based on previous knowledge of the field? If so, a non-bias approach would be much preferred for this kind of study to limit bias, identify new mechanisms, or confirm previous data. 

To clarify this issue, we have now added a paragraph to justify the use of ANN (lines 93-99), as well as an addition in the Methods section, where we indicate that this is the usual protocol for the application of our technology (TPMS) (lines 503-520).
In turn, 
(i) the number of proteins (=14) makes it impossible to carry out other ‘non-biased’ analyzes such as an enrichment, to allow the identification of new mechanisms associated with the disease; 
i) the aim of our study was to identify potential biomarkers. As such, a strategy with "bias" towards what is known in the disease allows, from a wide list of proteins, to select those candidates that are most likely to have an actual relationship with the disease.

1.     Line 409 – 422 should probably be in the results as it would give context to the reader and help understand the direction taken by the authors. 

Done. Thanks for catching this.

4.     “we prioritized 14 protein candidates” – How did the authors make the decisions to take these proteins – What were the thought process? 

Given that we have restructured/rewritten the Methods and Results section, this sentence has now been removed and the process that we carried out is better contextualized
In addition, we have now applied slightly different nomenclature criteria, in order to answer questions 11 and 11.1. below (see below for more details). Thus, the 14 proteins are now labelled as ‘the most differentially expressed proteins’ (rather than as ‘candidates’). These 14 proteins come from selecting the iTRAQ proteins that meet the control/patient value ratio filter >1.676, which was identified considering proteins with a number of peptides >10. Please see the revised manuscript text for clarification.

5.     “least a two-fold higher change in one of the control pools than in the patient 113 pool” – What is patient 113 pool? The changes were only seen in one pool and not all of them? Why was this specific pool chosen rather than others or testing all of them? 

We have rephrased this part to make it more suitable to the nomenclature of the technique. We have made an introduction in the methods part 4.2.5 in order to explain this further (lines 503-520).

9. The data in the ANN scores for most, if not all the proteins as written in table 3 do not match the ANN score of Supp Table 3. For example, MYH1 as ANN score of 93 in table 3 and 91 in supp table 3. This actually changes the category of a few of these proteins, hence disrupt the reason for selection of a few targets for WB confirmation.

Thank you very much for your insightful comment. 
Through machine learning and pattern recognition techniques, TPMS-based artificial neural networks (ANN) “learn” how areas of the human proteome are functionally related, using drug-indications and drug-adverse drug reactions relationships over the human protein network. The evaluation of the ANN allows to obtain predictive functional relationship scores for pairs of protein sets, and considers each protein set as a complete definition. In our study, former Table 3 (i.e., new Table 5 in the revised version of the manuscript) shows the results obtained when evaluating the relationship of each candidate with respect to the complete definition of GSDV (McArdle disease). However, as GSDV was defined by several (and functionally different) motives, we aimed to explore whether our ANN could detect relationships between candidates and individual motives that could be missed when considering the complete GSDV definition, which is what is actually shown in Suppl. Table 3. Thus, while we would expect that candidates with high scores in the general definition of GSDV would have at least one high ANN score when evaluating the motives (or vice versa), this is not necessarily true as per the nature of the model. Accordingly, the appearance of different ANN scores in former Table 3 (now Table 5) and Suppl Table 3 is normal. Some minimal changes/additions have been added in the methodology (lines 503-520) and results sections (lines 176-226) to clarify this issue, as well as to clearly state the criteria for selection of candidates for WB confirmation.

10. First I want to commend the authors in providing the full WB, as this should be standard for publication. However, I have some concerns with the presentation. There is appalling variability in alpha-tubulin and loading altogether despite having known protein loaded. This is also seen in targets.
1.     I am afraid that “Upper panel: Representative blots” is quite a stretch – all the lanes should be shown to show variability.

Thank you for your comment. We believe the variability in the figure is evident since the alpha-tubulin bands intensities display the variability of the WB. Our aim was to show the results of the WB more clearly to the readers using only a figure displaying an individual blot for each of the candidate proteins. In addition, we have now included a supplementary figure (Suppl. Figure 1) that displays all the blots for each patient and control. Accordingly, we have minor modified the text (line 274) and the Figure 2 legend (former Figure 1, has become Figure 2 in the revised manuscript (lines 280-281).

2.     In the same way, a dot plot should be used to show the spread of the data.

We have now changed the bottom panel of Figure 2 (Former Figure 1) to a dot-plot format to show the ratios of each individual (patients and controls) for each protein.

3.     Pooling quantification of two membranes (WB A and WB B) is extremely risky and is arguably not a sound procedure considering that the two membranes are likely not exposed to the same environmental factors, conditions, etc.

To minimize the variability due to experimental conditions, we performed the electrophoresis in two parallel gels (i.e. corresponding to WB A and WB B) that were running in the same electrophoretic cuvette and conditions. We also carried out the transfer of the separated products to the membranes, as well as the incubation with primary and secondary antibodies using the same conditions for WB A y WB B. Exposure of membranes was done consecutively.

4.     Many of the blots are overexposed (at least many lanes are), likely falling outside of the semi-quantitative range of chemilescence.

Thank you for your comment.
In our experience, homogenates from human muscle biopsies display very high inter-individual variability both when we analyzed testing proteins or loading control proteins; and consequently, we use ratio values of the testing protein and the loading control to perform statistical comparisons.

Thus, we set up each western blot corresponding to each protein in order to keep the linear range of the testing protein in combination with the linear range of the load control protein. As the reviewer indicates, it is true that it may seem in Figure 2 (former Figure 1) that some of the bands are apparently overexposed -saturated- but that is not the case, since we were very careful to obtain the proper exposure for each protein with that of its corresponding load control. 

5.     With such variation on the blot, I am surprised to see a reasonable error bar. It is not clear to me whether the data represent mean + SD or SEM. This should be written in the figure caption along with the N-number for each experiment.

Figure 2 (former Figure 1) now show dot-plots and embedded dispersion lines to show mean and SD. This is now indicated in the Figure legend (the dot of each individual = mean of two quantifications; i.e. two technical replicates) (lines 282-283)

6.     Explain the reasoning for choosing Mann-Whitney test rather than Welsch or regular t-test?

We used Mann-Whitney test as a non-parametric test, instead of using parametric tests (e.g. t-test) for comparisons due to the low sample size in each group (patients and controls) as well as to avoid bias attributed to the assumption of normal distribution of the data in each group.

11. Figure 2. “Symbols indicate the pathophysiological motive in which the protein is involved. No candidates are in purple color, and candidates are highlighted according to a colored gradient depending of their probability of being candidate as indicated in the candidate ratio scale.” - This statement is very confusing especially how candidate is used throughout the manuscript.
1.     Candidates … depending of their probability of being candidate – What does this mean exactly? Candidate of what? Please refine or use an alternative word to ensure clarity.

We have now restructured the candidate ratio scale to adapt it to the changes/additions we made in the manuscript, as well as for clarity purposes (Figure 1). In addition, we have now used throughout the entire text the following nomenclature criteria:
- ‘The most differentially expressed proteins’: proteins obtained from the statistical analysis in two phases (identification of threshold control/patient values ratio and application of the threshold to the 178 proteins), and mapped to SwissProtKB (revised UniProt), which were 14.
- ‘Candidate proteins’: proteins that have given a signal in the ANN analysis (or are selected by prior knowledge, i.e. ACTN3) and were validated by western blot.

2.     No candidate are in purple color – Please clarify.

We have revised the colors and modified the Figure 2 (former Figure 1), accordingly.

12. Statement: “our findings suggest that decreased expression of the aforementioned proteins in GSDV could explain the altered muscle contractile function and a probable alteration of muscle calcium kinetics in this disorder.” While this is true, reduction of protein does not necessarily entail functional deficits and functional experiments are required - The authors may consider discussing this aspect.

We agree. We have now toned down our statement (while on the other hand avoiding further speculation), and the sentence has been rewritten as follows: 
our findings suggest that decreased expression of the aforementioned proteins in GSDV could be associated, at least in part, with the altered muscle contractile function and a probable alteration of muscle calcium kinetics in this disorder (line 340).

Minor comments

1. How was the protein interactome obtained (software would be helpful)? This can be briefly mentioned in the results to help reader understand steps.

The protein interactome was drawn from the human protein network applied for building TPMS technology-based models (using publicly available sources), as described in the methodology section. Open-source network visualization software Cytoscape was used for building the interactome around the candidates (line 259, Figure 2 legend).
We completely agree with the Reviewer that we might have overlooked the clarity of the sources for the interactome, and that we should have stated the use of the visualization software. This information has now been included in the manuscript, as well as a short explanation of the source of the interactome in the results section (lines 205-208) and in Methods section 4.3.1. (lines 531-540).

2. Table 4 Define Interactor GSDV effector genes

Done (in new Table 5, since former Tables 3 and 4 have been combined in new Table 5). 

3. Statement 261-262 with comparison of ALS if perhaps not a fair and appropriate comparison given largely different pathogenic process and phenotype. ALS is highlighted by chronic denervation, muscle atrophy (and muscle weakness, rather than sustained muscle contraction), likely leading to reduced amount of these proteins. To my understanding, muscle atrophy is highlighted in the proteomic screen and as such the comparison is relatively weak.

We agree. We have simply removed this former sentence.

On behalf of all coauthors, many thanks for this insightful review.

Round 2

Reviewer 2 Report

As mentioned in my former review, the highly important question if the upreguated proteins ar primarily or secondarily upregulated is not adressed with an adequate control group with other forms od muscle cramping. A normal control group can not adress this issue.

Author Response

Comment appreciated.

Of note, the correct term in the context of McArdle disease is not ‘muscle cramps’, which refers to an acute phenomenon, but rather ‘muscle contractures’, a phenomenon that is usually also triggered by exercise (or sometimes by other stressors such as anxiety crises) but can persist during hours or days/weeks. Both phenomena are quite different from a physiological and electrophysiological point of view, as reflected by the EMG signal they elicit (i.e., acute, ‘epilepsy’-like discharges vs. normal, respectively). In fact, McArdle patients are not particularly prone to suffer cramps per se during exercise (although they of course can suffer cramps like anyone else) and for this reason, we have not used the word ‘cramp’ at any point of the manuscript when speaking about McArdle disease specifically.

On the other hand, we are not really aware of any other neuromuscular condition characterized by such frequent episodes of long-lasting contractures as McArdle disease (which represents ‘the paradigm of exercise intolerance’) that could be used as an ‘unhealthy’ control in order to discern if the differences we found in protein regulation in the patients’ muscles were ‘primary’ or ‘secondary’ in nature. McArdle disease is indeed a quite unique entity per se. In turn, some other neuromuscular conditions (e.g., CPT II deficiency) can be associated with frequent episodes of persistent muscle damage (which is a phenomenon different from contractures and is reflected by high serum CK levels, as explained in the manuscript).

Anyway, we have emphasized at the end of the Discussion section the limitation of not having used as a control group, patients with other neuromuscular conditions associated with muscle contractures (lines 370-375). In this regard, we would kindly request the Reviewer as well as the Editor to understand how difficult and costly it would be to run all the experiments again with such an additional group, which in any case would be quite difficult to gather. Indeed, an ‘ideal’ control group would be one composed of patients with other glycogenosis associated with muscle signs (i.e., phosphofructokinase deficiency, also known as glycogenosis type VII or Tarui disease), but we hardly know any patient with this condition in Spain. And, to the best of our knowledge, a control group of healthy people is always a good (and in fact necessary) reference in any study. We have also made it clear in the manuscript (starting from the Abstract section) that the controls were healthy people free of McArdle-like features.

In summary, we hope that the vast amount of honest effort we have put into this manuscript as well as the novel data we report can compensate for the aforementioned limitation of a lack of an additional study group

Reviewer 3 Report

The authors have done great work in answering the comments and concerns. I am pleased with their revisions. They have significantly improved their manuscript.  

I would recommend that this response is featured (see below in Italics) in the manuscript's methods. This is an important detail that may be negatively seen by readers, but the authors have taken the right approach to ensure consistency and reduced bias.

3.     Pooling quantification of two membranes (WB A and WB B) is extremely risky and is arguably not a sound procedure considering that the two membranes are likely not exposed to the same environmental factors, conditions, etc.

To minimize the variability due to experimental conditions, we performed the electrophoresis in two parallel gels (i.e. corresponding to WB A and WB B) that were running in the same electrophoretic cuvette and conditions. We also carried out the transfer of the separated products to the membranes, as well as the incubation with primary and secondary antibodies using the same conditions for WB A y WB B. Exposure of membranes was done consecutively.

I also noted one small typo - line 297 - There is '')'' that does not belong there at the end of the sentence.

All the best to the authors and their next endeavours

Author Response

Dear Reviewer,

Comments appreciated.

Following your recommendation, we have included a paragraph in the Methods section (Lines 583-588); and corrected the typo “)” (Line 311).

Round 3

Reviewer 3 Report

All comments have been answered appropriately